# Cardioprotective Effect of Low Level of LDL Cholesterol on Perioperative Myocardial Injury in Off-Pump Coronary Artery Bypass Grafting

**DOI:** 10.3390/medicina57090875

**Published:** 2021-08-26

**Authors:** Tomasz Kamil Urbanowicz, Anna Olasińska-Wiśniewska, Michał Michalak, Aleksandra Gąsecka, Michał Rodzki, Bartłomiej Perek, Marek Jemielity

**Affiliations:** 1Cardiac Surgery and Transplantology Department, Poznan University of Medical Sciences, 61-848 Poznań, Poland; annaolasinska@ump.edu.pl (A.O.-W.); michal.rodzki@skpp.edu.pl (M.R.); bperek@ump.edu.pl (B.P.); mjemielity@poczta.onet.pl (M.J.); 2Department of Computer Science and Statistics, Poznan University of Medical Sciences, 60-529 Poznań, Poland; michal@ump.edu.pl; 31st Chair and Department of Cardiology, Medical University of Warsaw, 02-106 Warsaw, Poland; gaseckaa@gmail.com

**Keywords:** cardiac surgery, cardiovascular disease, prognosis, therapy, drugs

## Abstract

*Background and Objectives*: Coronary artery disease is still a major cause of death in developed countries. Low-density lipoprotein cholesterol (LDL-C) lowering with statin therapy is a key strategy in major acute coronary events’ prevention. The aim of the study was to establish if there is a cardioprotective effect of pre-operative LDL lowering therapy on perioperative myocaridal injury in patients undergoing off-pump coronary artery bypass grafting (CABG). Moreover, the impact of pre-operative LDL level on long term outcome was analysed. *Materials and Methods*: The retrospective single center analysis included 662 consecutive patients (431 (65%) males and 231 (35%) female, mean age of 65 ± 8) referred for cardiac surgery due to stable chronic coronary syndrome between 2012–2018. The follow up was 9 years. *Results*: A statistically significant difference was found in postoperative serum Troponin-I for LDL thresholds of 1.8 mmol/L (*p* = 0.009), 2.6 mmol/L (*p* = 0.03) and 3.0 mmol/L (*p* = 0.001). The results indicate that cardioprotective role of LDL is achieved within LDL concentration rate below 1.8 mmol/L (<70 mg/dL). Five patients died perioperatively, whereas 1-year and 9-year overall mortality rates were 4% (*n* = 28) and 18.6% (*n* = 123), respectively. Comparing the survival group with diseased, Mann-Whitney U test showed a statistically significant difference in HDL-C (*p* = 0.007), Troponin (*p* = 0.009), Castelli index (*p* = 0.001) and atherogenic index (*p* = 0.004). Preoperative levels of total cholesterol, LDL-C and HDL-C did not significantly differ between survivors and diseased. The 9-year mortality risk did not differ significantly between subgroups divided according to LDL-C thresholds of 1.4 mmol/L (55 mg/dL), 1.8 mmol/L (70 mg/dL), 2.6 mmol/L (100 mg/dL) and 3.0 mmol/L (116 mg/dL). *Conclusions:* Preoperative low level of LDL-C cholesterol (below 1.83 mmol/L, 70 mg/dL) has a cardioprotective effect on perioperative myocardial injury in off-pump coronary artery bypass grafting.

## 1. Introduction

Coronary artery disease is still a major cause of death in developed countries. Genetic and environmental factors combined with lifestyle pose a risk for the development of atherosclerotic disease. Complex coronary disease can be treated with optimal medical therapy and percutaneous or surgical revascularization. The off-pump surgery is a safe option due to its relatively low complications rate [1,2].

Low-density lipoprotein cholesterol (LDL-C) lowering with statin therapy is a key strategy in acute and chronic coronary syndromes’ prevention according to randomized clinical studies and high-volume meta-analyses [3,4,5]. Problems with patients’ noncompliance may be overcome with good meticulous argumentation supported by scientific evidence [6]. Therefore, studies which prove the cardioprotective role of LDL lowering are crucial [7].

The protective effect of pretreatment statin therapy on periprocedural myocardial damage was presented in several large trials [8,9,10]. The surgical intervention may be related to some defined complications, including the amount of myocardial injury [11]. Preoperative statin therapy is suspected to improve post-operative results, but in several studies it did not influence the risk of either perioperative acute kidney dysfunction or long-term graft patency [12,13].

The aim of the study was to establish if there is a cardioprotective effect of pre-operative LDL lowering therapy on perioperative myocardial injury in patients undergoing off-pump coronary artery bypass grafting (CABG). Moreover, the impact of pre-operative LDL level on long term outcome was analysed.

## 2. Materials and Methods

The study was based on retrospective analysis of 662 consecutive patients including 431 (65%) males and 231 (35%) females of mean age of 65 ± 8 referred for cardiac surgery to our department due to stable chronic coronary syndrome between 2012–2018. Patients qualified for cardiac surgery based on coronary angiography results, with 417 (63%) and 245 (37%) presenting three vessel and left main (LM) disease. The concomitant diseases included: arterial hypertension in 477 (72%), diabetes mellitus in 218 (33%), chronic obstructive pulmonary disease 120 (18%) and peripheral artery disease in 93 (14%) patients. All patients were treated with lipid lowering agents including simvastatin (*n* = 203), atorvastatin (*n* = 236), and rosuvastatin (*n* = 223). Demographical and clinical data are outlined in Table 1.

All the procedures were performed via complete median sternotomy without cardiopulmonary bypass as an off-pump coronary artery bypass grafting (OPCAB) technique.

On the day of admission, blood samples including serum lipid profile analysis were collected. The Castelli indices 1 (total serum cholesterol/HDL-C) and 2 (LDL-C/HDL-C) were calculated [14]. During the postoperative time, the serum troponin-I (Trop-I) levels and kinase creatinine isoenzyme (CK-MB) were collected. The preoperative LDL-C levels were compared with myocardial injury markers including Troponin-I serum levels. We subdivided the presented group into subgroups representing the following thresholds of LDL-C according to ECS guidelines: below 1.43 mmol/L (55 mg/dL), below 1.83 mmol/L (70 mg/dL), below 2.59 mmol/L (100 mg/dL) and below 3.0 mmol/L (116 mg/dL) [15].

The median follow up was 5.5 ± 2.6 years and up to 9 years (2012–2021) and included analysis of all-cause mortality confirmed by the Polish National Health Service database.

All patients signed written informed consent for routine surgery. The study received positive agreement from the Poznan University of Medical Sciences’ Ethics Committee (No 55/20 from 16 January 2020).

### Statistical Analysis

Continuous variables were reported as mean ± standard deviation (SD) if data followed the normal distribution (Shapiro-Wilk test), otherwise the data were presented as median and interquartile range (Q_1_–Q_3_). Categorical data were presented as numbers and percentages. The comparison of interval parameters between survivors and deceased group was performed by Mann-Whitney. The max troponin I level between groups denoted according to ECS cardiovascular LDL-C threshold was analyzed by Kruskal-Wallis and Dunn’s post-hoc test. Categorical data were analyzed by chi-square test for independence. The Cox’s proportional hazard regression model was used to check if analyzed demographical and clinical data could be a risk factor for all-cause death. The analysis was performed with the use of statistical package TIBCO Software Inc., Palo Alto, CA, USA (2017). Statistica (data analysis software system), version 13. http://statistica.io (Since 2000, site license). All tests were considered significant at *p* < 0.05.

## 3. Results

All patients admitted for surgical procedure were on statin therapy. The precise data is presented above Table 1. The mean values of preoperative serum cholesterol, LDL-C, high-density lipoprotein cholesterol (HDL-C) levels are presented in Table 1. All patients underwent off-pump coronary artery bypass grafting with mean skin-to-skin time of 142 ± 41 min and mean hospitalization time was 11 ± 4 days.

### 3.1. Myocardial Injury

We compared the postoperative myocardial injury markers between LDL-C subgroups. Mann-Whitney test did not reveal significant statistical difference between max postoperative Troponin-I and subgroup of patients with LDL-C results below 1.4 mmol/L (*p* = 0.08). There was a statistically significant difference noticed in postoperative serum Troponin-I for LDL-C thresholds of 1.8 mmol/L (*p* = 0.009), 2.6 mmol/L (*p* = 0.03) and 3.0 mmol/L (*p* = 0.001). The multiple comparison p values comparing postoperative Troponin-I and LDL-C values are presented in Table 2. The results indicate that cardioprotective role of LDL-C is achieved within LDL-C concentration rates below 1.8 mmol/L (<70 mg/dL).

### 3.2. Perioperative and Long-Term Survival

There were five (0.8%) perioperative deaths in the presented group and 4% (*n* = 28) overall 1-year mortality. There were 13 (2%) postoperative excessive bleedings requiring reoperation. In the 9-year follow-up overall mortality rate was 18.6% (*n* = 123).

Comparing the survival group with diseased, Mann-Whitney U test found statistically significant difference in HDL-C (*p =* 0.007), Troponin (*p =* 0.009), Castelli index (*p =* 0.001) and atherogenic index (*p =* 0.004). Preoperative levels of total cholesterol, LDL-C and HDL-C did not significantly differ between survivors and diseased. Despite significant difference between both groups, the Cox regression analysis did not confirm any of the presented factors as a significantly predictive.

We subdivided the study group into subgroups representing the following thresholds of LDL-C according to ECS guidelines: below 1.4 mmol/L (55 mg/dL), below 1.8 mmol/L (70 mg/dL), below 2.6 mmol/L (100 mg/dL) and below 3.0 mmol/L (116 mg/dL) and analyzed the 9-year mortality rates (Figure 1).

First, we evaluated patients below and above the threshold of 1.4 mmol/L, and then 1.8 mmol/L. Finally, we compared mortality rates in subgroups divided according to LDL-C ranges (Table 3).

There were 128 (20%) and 516 (80%) patients with LDL-C below and above 1.4 mmol/L. The mortality in the presented subgroup was 19 (14.8%) vs 104 (20.2%), respectively. The difference was not statistically significant. The number of patients presenting values below the threshold of 1.8 mmol/L was 233 (36%), and 411 above (64%). The mortality risk in presented subgroup was 15.5% (*n* = 36) vs 21.2% (*n* = 87), respectively. The difference was not statistically significant.

The study group was divided into subgroups representing the following thresholds of LDL-C according to ECS guidelines [15]: 1.4 mmol/L (55 mg/dL), 1.8 mmol/L (70 mg/dL), 2.6 mmol/L (100 mg/dL) and 3.0 mmol/L (116 mg/dL). We compared (Table 3) consecutive subgroups in terms of nine-year mortality, but these results also were not statistically significant.

## 4. Discussion

The results of our study present the LDL-C values that may have a cardioprotective role during OPCAB procedures. A significant relation between preoperative LDL-C values and postoperative Troponin-I release was observed. The LDL-C below 1.83 mmol/L (<70 mg/dL) characterized the patients with significant reduction of postoperative Troponin-I release. Therefore, we proved that pre-operative LDL lowering is important for peri-operative management. Statin therapy may protect the heart muscle from the intra-operative and post-operative injury or at least diminish it. The protection may result from pleiotropic effect of statins, including anti-inflammatory effect and plaque stabilization. Similar results were presented in the meta-analysis of Farkouh et al. regarding percutaneous coronary interventions (PCI) [16]. Navarese et al. also pointed out the significance of intensive statin therapy on cardiovascular risk reduction [17].

Statins’ role in atherosclerotic plaque stabilization and regression is already known. The EASY-FIT study presented the statin effect on fibrous cap thickness, suggesting that LDL-C < 70 mg/dL stabilizes coronary atherosclerotic plaques [18]. The use of lipid-lowering medications may slow down the rate of atherosclerosis disease progression, as was shown in the ASTERIOD trial [19]. The inflammatory concept in the pathogenesis of atherosclerosis recognizes atherogenesis as an active process with mononuclear phagocytes’ contribution in all stages of this disease [20]. The beneficial effects of statins relate to their anti-inflammatory properties such as reduction of T-cell activity, chemokines, cytokines, and adhesion molecules including ICAM-1, lymphocyte function-associated antigen-1, monocyte chemotactic protein-1 and Th1-type chemokine receptors on T cells’ inhibition [21].

In the multicenter REVERSAL trial, Nissen et al. reported reduced progression of coronary atherosclerosis by change in percent atheroma volume achieved by statin therapy [22]. The SATURN trial proved regression of atherosclerotic plaque in patients receiving high-intensity statin therapy [23]. Interestingly, due to individual plaque anatomy and individualized systemic biological effect of statins, the optimal dosage may be difficult to determine [24]. Recently, a very aggressive LDL-C reduction was shown to decrease the risk for cardiovascular events and improve survival with a threshold as low as 40 mg/dL in the REDUCE-IT trial [25].

Cardioprotective role of statin therapy during PCI procedures is related to myocardial perfusion improvement resulting in myocardial infarction risk reduction [26]. Marenzi et al. presented the cardioprotective role of long-term statin therapy on infarct size in patients with ST elevation myocardial infarction treated with PCI in multivariable analysis [27]. Auguardo et al. proved statin influence on Troponin release after PCI as an independent myocardial protection predictor [28].

Troponin release after PCI does not straightforwardly predict all-cause mortality according to Lippi et al. [29]. Isolated myocardial injury diagnosis based only on Troponin-I release does not translate into unfavourable prognosis [30]. On the contrary, in meta-analysis performed by Nienhuis et al., nonfatal myocardial infarction and mortality risk were increased in patients with post-procedural troponin elevation [31]. In surgical revascularization, the study of Wang et al. confirmed that combination of Troponin release and electrocardiographic changes with echocardiographic ejection fraction deterioration was a mortality predictor in multivariable analysis, contrary to sole Troponin release [32]. Undoubtedly, Troponin is the most reliable myocardial injury marker [33]. In the study of Machado et al. study and the meta-analysis of Buse et al., the Troponin release predicted 30-day and mid-term mortality following coronary artery bypass grafting, respectively [34,35]. The comparison of these studies support the importance of Troponin release. This may explain the results of our study regarding lack of predictive Troponin values in the long-term, 9-year observation time.

Surprisingly, the pre-operative LDL level did not correspond with long term mortality. Navarese et al. in their publication presenting meta-analyses and meta-regressions, pointed out that more intensive LDL-C lowering compared with less intensive strategy was associated with a greater reduction in risk of total and cardiovascular mortality in trials of patients with higher baseline LDL-C levels (100 mg/dL) [17]. Therefore, it might be expected that, in our study, long term survival would be improved in patients with lower baseline LDL-C levels. However, we must point out that our study reflects a real-life registry, without a strict researchers’ influence on patients’ compliance, including patients with different types of statin used before and after surgery in terms of duration, type and dosage. Though this may be treated as a limitation, these observation incline towards even more strict control of statin use in patients with coronary artery disease treated with coronary artery bypass grafting, both before and after the procedure.

## 5. Conclusions

Low level of LDL cholesterol (below 1.8 mmol/L, 70 mg/dL) has a cardioprotective effect on perioperative myocardial injury in off-pump coronary artery bypass grafting.

## Figures and Tables

**Figure 1 medicina-57-00875-f001:**
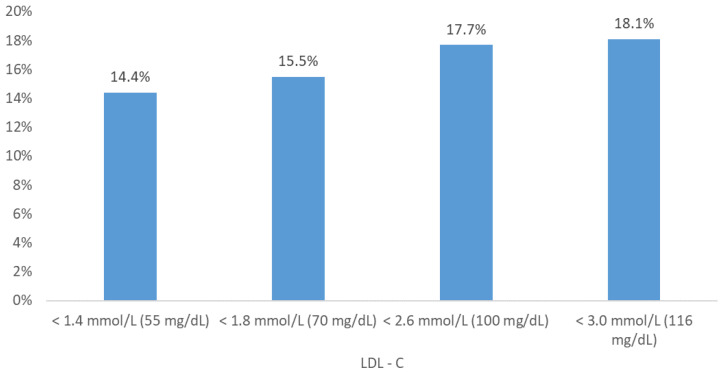
Nine-year cumulative mortality in subgroups divided according to ECS cardiovascular LDL-C threshold.

**Table 1 medicina-57-00875-t001:** Demographical and clinical data.

Parameter	No-662
Age (year) (mean, SD)	65 ± 8
Gender (male/female) (*n*, %)	431 (65%)/231 (35%)
Indication for surgery: - Left main stenosis (*n*, %) - 3 vessels disease (*n*, %)	245 (37%) 417 (63%)
Concomitant diseases: - Arterial hypertension (*n*, %) - Diabetes mellitus (*n*, %) - COPD (*n*, %) - Peripheral artery disease (*n*, %)	477 (72%) 218 (33%) 120 (18%) 93 (14%)
Preoperative labolatory parameters: - Cholesterol (mmol/L) ((median, Q1–Q3) - HDL-C (mmol/L) (median, Q1–Q3) - LDL-C (mmol/L) (median, Q1–Q3) - Rastelli index (median, Q1–Q3) - Atherogenic index (median, Q1–Q3)	4.0 (3.42–4.77) 1.27 (1.06–1.5) 2.1 (1.6–2.7) 3.18 (2.6–3.9) 1.7 (1.2–2.2)
Postoperative myocardial injury parameters: - max Troponin I (ng/mL) (median, Q1–Q3) - max Ck-MB (ng/mL) (median, Q1–Q3)	2.43 (0.95–5.96) 7.6 (2.93–14.4)
Observation time (years) (mean, SD)	5.5 ± 2.6

Abbreviations: COPD—chronic obstructive pulmonary disease, HDL-C—high-density lipoprotein cholesterol, LDL-C—low-density lipoprotein cholesterol, max—maximum, Q—quartile.

**Table 2 medicina-57-00875-t002:** Kruskal-Wallis and Dunn’s post-hoc test results for max Troponin-I serum level vs LDL-C.

	Kruskal-Wallis Test: H (4, *n* = 640) = 13.149 *p* = 0.0106			
	LDL-C below1.43	LDL-C 1.43–1.82	LDL-C 1.82–2.59	LDL-C 2.59–3.0
LDL-C below1.4 mmol/L				
LDL-C 1.4–1.8 mmol/L	1.0000			
LDL-C 1.8–2.6 mmol/L	1.0000	1.0000		
LDL-C 2.6–3.0 mmol/L	1.0000	1.0000	1.0000	
LDL-C above 3.0 mmol/L	0.0166	0.0294	0.3961	0.2168

Abbreviations: LDL-C–low-density lipoprotein cholesterol.

**Table 3 medicina-57-00875-t003:** Nine-year mortality in subgroups divided according to ECS cardiovascular LDL-C threshold.

LDL-C Level	Mortality in the Subgroups *n* (%)	Total Number of Patients in the Subgroup
below 1.4 mmol/L	19 (14.8%)	128
1.4–1.8 mmol/L	17 (16.2%)	105
1.8–2.6 mmol/L	43 (20.2%)	213
2.6–3.0 mmol/L	16 (20.5%)	78
above 3.0 mmol/L	28 (23.3%)	120
Chi-square *p*-value = 0.4435		

Abbreviations: LDL-C—low-density lipoprotein cholesterol.

## Data Availability

All data will be available under correspondence e-mail address for 3 years following the publication after request that would be justifiable.

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
