# Peer review of "Cardioprotective Effect of Low Level of LDL Cholesterol on Perioperative Myocardial Injury in Off-Pump Coronary Artery Bypass Grafting"

_medicina, 2021, doi:10.3390/medicina57090875_

Round 1
Reviewer 1 Report
The science presented is interesting, results and conclusion are clear. However, the text requires minor language corrections:
Page 2 line 75 - please correct ‘coornary’ to ‘coronary’
Page 2 line 77 - ‘overcame’ should be ‘overcome’ – it doesn't sound the sentence is in the past but just a routinely problem that can be avoided.
Page 2 line 83 - ‘ defind’ should be ‘defined’ – ‘amunt’ should be ‘amount’
Page 2 line 88 – ‘myocaridal’ please correct to ‘myocardial’
Page 3 line 98 – please change ‘chronic obstructive airway disease’ to ‘chronic obstructive pulmonary disease’
Page 3 line 94 – the authors are stating that patients were included between 2012 and 2018, and page 4 line 128 mentions that the patients were followed for a 9 years follow up. I would advise to reword the latter statement to ‘The median follow up was 5.5 ± 2.6 years and up to 9 years’
Page 8 line 287 – please change ‘in contrary’ with ‘on the contrary’.
Author Response
Dear Reviewer,
Thank you for your valuable comments.
All the corrections have been made according to your suggestions.
Kind regards
Tomasz Urbanowicz
on behalf of all co-authors
Reviewer 2 Report
This is retrospective single center analysis on Cardioprotective effect of low level of LDL cholesterol on perioperative myocardial injury.
The findings of this study are not new discovery . This is well known facts of cardioprotective action with targeted low LDL-C by numbers of trials and well established guidelines (eg. ESC, ACC, ect) .
The authors are suggested to describe exactly on which year ESC guidelines were used as ESC guidelines have usually been updated year by year .
The study didn't describe clearly on the participants taking which type of lipid lowering agents and dosage . Advise to describe more data on that points .
Author Response
Dear Reviewer,
Thank you for your valuable comments.
We strongly agree that cardioprotective action of low LDL is well known. Our main finding which we want to underline is its cardioprotective value after CABG shown by troponin level difference between study subgroups.
We used latest 2019 ESC guidelines (Reference No 15).
The lipid lowering drugs were added into the manuscript including 3 different types of them.
Kind regards
Tomasz Urbanowicz
on behalf of all co-authors